# 2′-Fucosyllactose Ameliorates Oxidative Stress Damage in d-Galactose-Induced Aging Mice by Regulating Gut Microbiota and AMPK/SIRT1/FOXO1 Pathway

**DOI:** 10.3390/foods11020151

**Published:** 2022-01-07

**Authors:** Jin Wang, Jia-Qiang Hu, Yu-Jie Song, Jia Yin, Yuan-Yi-Fei Wang, Bo Peng, Bo-Wei Zhang, Jing-Min Liu, Lu Dong, Shuo Wang

**Affiliations:** Tianjin Key Laboratory of Food Science and Health, School of Medicine, Nankai University, Tianjin 300350, China; wangjin@nankai.edu.cn (J.W.); heric@mail.nankai.edu.cn (J.-Q.H.); yjsongqcyy@163.com (Y.-J.S.); spyinjia@163.com (J.Y.); wangyyf163@163.com (Y.-Y.-F.W.); iq1226jsnpb@hotmail.com (B.P.); bwzhang@nankai.edu.cn (B.-W.Z.); liujingmin@nankai.edu.cn (J.-M.L.); donglu@nankai.edu.cn (L.D.)

**Keywords:** 2′-fucosyllactose, aging, SIRT1, gut microbiota

## Abstract

The imbalance of reactive oxygen species is the main cause in aging, accompanied by oxidative stress. As the most abundant in human milk oligosaccharides (HMOs), 2′-Fucosyllactose (2′-FL) has been confirmed to have great properties in immunity regulation and anti-inflammatory. The research on 2′-FL is focused on infants currently, while there is no related report of 2′-FL for the elderly. A d-galactose-induced accelerated aging model was established to explore the protective effect of 2′-FL on the intestines and brain in mice. In this study, 2′-FL significantly reduced oxidative stress damage and inflammation in the intestines of aging mice, potentially by regulating the sirtuin1 (SIRT1)-related and nuclear factor E2-related factor 2 (Nrf2) pathways. In addition, 2′-FL significantly improved the gut mucosal barrier function and increased the content of short-chain fatty acids (SCFAs) in the intestine. The gut microbiota analysis indicated that 2′-FL mainly increased the abundance of probiotics like *Akkermansia* in aging mice. Moreover, 2′-FL significantly inhibited apoptosis in the brains of aging mice, also increasing the expression of SIRT1. These findings provided a basis for learning the benefits of 2′-FL in the aging process.

## 1. Introduction

Aging is a complex physiological process, leading to a gradual decline in physical health [1]. Under normal conditions, the content of reactive oxide species (ROS) is maintained at a relatively stable level. However, the activity of antioxidant enzymes that clears reactive oxygen species decreases as age increases, which results in a large accumulation of ROS in cells, impairing cell functions and accelerating the occurrence of aging [2]. The aging process is usually accompanied by the destruction of the immune function, causing various inflammatory reactions and cell apoptosis [3]. Aging is related to many factors, including energy metabolism, oxidative stress, cell apoptosis, and inflammation [4,5]. Adenosine 5′-monophosphate-activated protein kinase (AMPK) is a key molecule in regulating biological energy metabolism, including an α-catalytic subunit, a β-regulatory subunit, and a γ-regulatory subunit [6]. AMPK is an important molecule involved in the regulation of aging, which could activate sirtuin1 (SIRT1), forkhead box protein O1(FOXO1), peroxisome proliferator-activated receptor γcoactivator-1(PGC-1α), nuclear factor κB (NF-κB), and other downstream molecules to inhibit oxidative stress and inflammation [7]. SIRT1 is a member of the NAD^+^-dependent deacetylase family, considered as an important longevity gene. SIRT1 is involved in DNA repair, apoptosis, inflammation, aging, and other biological processes [8]. Nuclear factor E2-related factor 2 (Nrf2) is a vital antioxidant sensor related to the aging process caused by oxidative stress and can be activated by SIRT1 [9]. The activation of Nrf2 can further induce the transcription of target genes like heme oxygenase 1 (HO-1), protecting cells against oxidative damage [10].

Aging is closely related to changes in the environment of the intestine. A variety of bacteria are colonized in the intestine to form the gut microbiota. Aging decreases the diversity of gut microbes, like butyrate-producing species, and changes the composition, like a decrease in the abundance of *Firmicutes*/*Bacteroides* [11,12]. Short-chain fatty acids (SCFAs) are beneficial products metabolized by the gut microbiota, and their positive therapeutic effects on aging have been reported extensively in the literature [13,14]. The production of SCFAs is considered to connect the host and gut microbiota, noting the correlation between changes in the gut microbiota and the increase of frailty in the elderly. SCFAs regulate the gut microbiota, prevent the proliferation of pathogenic bacteria such as *Escherichia coli* by lowering the pH, and stimulate the growth of beneficial bacteria of *Firmicutes* [15,16]. In addition, the gut mucosal barrier is composed of the mucosal layer and the tight junction of intestinal epithelial cells, which facilitates the absorption of beneficial nutrients and prevents harmful substances and pathogens from passing through the epithelial cells [17]. Aging could cause damage to the gut mucosal barrier, which, in turn, causes systemic inflammation [18].

Aging usually affects the physiological functions of the brain, accompanied by various neurodegenerative diseases like Alzheimer’s disease. Studies have reported that excessive apoptosis activation is increasingly involved in several neurodegenerative diseases related to aging [19]. Moreover, the increase in ROS or the decrease in antioxidants could lead to cell apoptosis and promote the aging process [20]. Apoptosis has become an important factor in aging caused by free radicals.

As a natural prebiotic, 2′-fucosyllactose (2′-FL) is the trisaccharide with the highest content in human milk oligosaccharides (HMOs), accounting for 31% and constituting glucose, galactose, and fucose [21]. 2′-FL is neutral, has a low molecular weight, and is recognized as playing an extremely important role in host immunity. The unique nature of 2′-FL allows it to resist digestion in the gastrointestinal tract, with very little entering the blood and systemic circulation, causing the majority to easily reach the large intestine for microbial decomposition. At present, 2′-FL has achieved commercial production and has passed regulatory approvals in some countries. In recent years, the Food and Drug Administration has approved several companies to produce 2′-FL by chemical synthesis or microbial fermentation, as well as the European Union. 2′-FL has been rapidly developed as an ingredient used to supplement infant formulas, dietary supplements, and medical foods [22]. According to research reports, 2′-FL has various physiological effects, promoting the proliferation and colonization of intestinal probiotics like *Bifidobacterium*, inhibiting the reproduction of harmful bacteria [23,24], regulating the immune system, and reducing inflammation [25]. At present, there have been some literature reports on the effect of oligosaccharides in aging. For instance, acarbose improved the lifespan of aging HET3 mice in an in vivo study related to its special role in lowering blood sugar [26]. Chitosan oligosaccharides had obvious antiaging activity in an aging model, and its mechanism was related to antioxidation and immunity improvement in aging [27]. In addition, a synthetic prebiotic called galacto-oligosaccharides (GOS) was found to regulate the homeostasis of the aging intestinal tract by stimulating changes in the microbiome composition and host gene expression, specifically manifested in decreased intestinal permeability and increased mucosal production [28]. It is worth noting that GOS is currently one of the substitutes for HMOs, added to infant formula as an ingredient. In terms of functions such as regulating the gut microbiota and metabolites, 2′-FL has a better effect than GOS [29]. Another study on the growth and adhesion of *Streptococcus mutans* found that GOS promoted the growth of *Streptococcus mutans* as a carbon source, while 2′-FL could not. Both could reduce the adhesion of pathogenic bacteria, but the mode of inhibition might be different [30]. Previous studies in our laboratory confirmed that 2′-FL intervention could regulate the composition of the gut microbiota, and the colonization of *E. coli* O157 in the intestines of mice was decreased by over 90%. One of the potential reasons was that 2′-FL is a structural analog of intestinal mucosal glycoprotein, which acts as a bait receptor against pathogenic bacteria. [31]. Moreover, 2′-FL was reported to have a certain effect on promoting brain development [32]. In a population study, it was found that breastfeeding promoted the cognitive development of infants within 1 month, where 2′-FL was the main reason [33].

As a natural ingredient in breast milk, the research on 2′-FL is focused on infants currently. However, there is no related report of 2′-FL for the elderly. Based on the function of 2′-FL, we reasonably speculate that 2′-FL intervention can work potentially in treatment strategies for aging. Injections of low-dose d-galactose (d-gal) into mice were first reported in China for signs of accelerated aging. Neurological damage, decreased antioxidant enzyme activity, and a weakened immune response have been demonstrated in this model of aging. d-gal is a reducing sugar that reacts with free amino groups of protein and peptide amino acids to form advanced glycation end products. These biochemical processes mentioned cause oxidative stress and cellular damage. d-gal is converted to galactitol, which can produce reactive oxygen species in the cell and stimulate osmotic stress, as well as accumulate in the cell. This study will apply the d-galactose-induced accelerated aging model to investigate the potential mechanism of 2′-FL on the intestine and the effect on the repair of brain damage in aging.

## 2. Materials and Methods

### 2.1. Experimental Procedures of Animal

Male-specific pathogen-free (SPF) ICR mice (10 weeks old) purchased from Beijing Vital River Laboratory Animal Technology Co. Ltd. (Beijing, China) were raised and tested in accordance with the institutional guidelines. d-gal was used to build an accelerated aging model. Long-term injection of excessive d-gal could cause oxidative stress damage and inflammation, which led to aging and nerve dysfunction. This model was usually used for research on aging-related aspects.

In our study, mice were adapted to the environment with a standard diet for 1 week and then divided into 3 groups (12 mice in each group) that were the control group (Control), d-gal model group (d-gal), and 2′-FL intervention group (2′-FL). All animals were housed individually at 22–23 °C, 46–53% relative humidity with 12 h:12 h light–dark cycles. In the d-gal model and 2′-FL intervention groups, mice were injected with 300-mg/kg/day d-gal. Mice in the control group was injected with 0.9% saline with equal volume, relatively. The 2′-FL intervention was added to the feed at a 2.5-g/kg diet in the 2′-FL intervention group. The specific diet composition is listed in Appendix A. All diets were adjusted to the same energy level. After 8 weeks of diet, the mice were peacefully sacrificed after anesthesia. Blood was carefully collected and centrifuged (4 °C, 4000 rpm, 15 min) to obtain a serum, but also, the colon, colon contents, cecum contents, and brain were carefully collected. Serum and tissues were quickly transferred and stored at −80 °C for later analysis.

This animal experiment was ethically reviewed by the Institutional Animal Care and Use Committee of Nankai University, and the ethical approval number is 2021-SYDWLL-000472. All experimental procedures followed the guidelines of international animal welfare and ethics.

### 2.2. Biochemical Assays

Blood samples stored at −80 °C were used to measure oxidative stress indicators. Biochemical indexes were measured with commercial detection biochemical kits (Nanjing Jiancheng Bioengineering Institute, Nanjing, China), including superoxide dismutase (SOD), catalase (CAT), glutathione (GSH), malondialdehyde (MDA), and glutathione peroxidase (GSH-PX) in the serum.

### 2.3. Analysis of Quantitative Real-Time Polymerase Chain Reaction (qPCR)

qPCR was applied to determine the expression levels of genes in the colon and brain tissues. In the colon, the genes intestinal barrier-related, including mucoprotein 2 (MUC2), Claudin1, and E-cadherin; the genes inflammation-related, including tumor necrosis factor-α (TNF-α) and interleukin-1β (IL-1β); the genes oxidative stress-related, including Nrf2, Kelch-like ECH-associated protein 1 (Keap1), HO-1, and NADPH quinone oxidoreductase-1 (NQO-1); the genes of SCFAs receptors, including G-protein-coupled receptor 41 and 43 (GPR41 and GPR43); and the genes aging-related, including SIRT1, FOXO1, and PGC-1α, expression levels were determined by qPCR. In the brain, the genes apoptosis-related, including B-cell lymphoma-2 (Bcl-2), Bax, and caspase-3; the genes oxidative stress-related, including Nrf2, Keap1, HO-1, and NQO-1; and the genes aging-related, including SIRT1 and FOXO1, were measured for their expression levels by qPCR. The experimental methods are shown in the Appendix A, and the oligonucleotide primers are displayed in Appendix A.

### 2.4. Analysis of Western Blot

Western blotting was applied to determine the expression levels of proteins in the colon. We measured the protein expression levels of SIRT1, FOXO1, p-AMPK, and AMPK in the colon. The specific methods are shown in the Appendix A.

### 2.5. Immunocytochemistry

The intestinal tissue was made into paraffin sections with a thickness of 5 µm and stained for the expression of MUC2 in the mucosal layer. Immunohistochemistry was performed by the Servicebio Company (Wuhan, China), and the method was as follows: The sections were deparaffinized for 10 min, followed by 3% H_2_O_2_ soaking for 10 min. Then, a citric acid buffer was added to the sections and put in a microwave oven for 3 min. The serum was used to block the sections at 37 °C for half an hour and then incubated overnight at 4 °C with the primary antibody (1:500 dilution, GB11344, Servicebio). The mixture was incubated at 37 °C for half an hour with the secondary antibody (1:200 dilution, GB23303, Servicebio), followed by adding developers to develop the color. The color-developed film was rinsed with water for a while and later soaked in hematoxylin for half a minute. The counter-stained film was dehydrated and covered with a cover glass, sealed, and dried for microscopy.

### 2.6. Staining with Hematoxylin–Eosin (H&E) for Observation

The brain tissue was made into paraffin sections about 5 μm, which were stained with H&E, encapsulated as slides, and photographed under the microscope for analysis.

### 2.7. GC method for SCFA of Colon Contents

The fecal sample was lyophilized, its dry weight weighed, homogenized in pure water (1 mL), and then centrifuged (4 °C, 13,000 rpm, 10 min) to obtain the supernatant. The supernatant was added to 100 μL of 10% sulfuric acid and 0.5 mL of ether and vortexed for 2 min. Continuously, the mixture was extracted for 5 min under the shock. The extracted solution was centrifuged (4 °C, 13,000 rpm, 10 min) to take the above ether phase through 0.22-μm filter membranes. Finally, the ether extract was stored in a brown sample vial to determine the SCFA content. A DB-FFAP capillary column (30 m × 0.25 mm × 0.25 μm, Agilent) was used for determination in gas chromatography (7890A, Agilent, Santa Clara, CA, USA). Agilent’s MSD ChemStation (E.02.00.493) was adopted for data processing.

### 2.8. Gut Microbial Analysis of Cecal Contents

Bacterial DNA extraction, PCR amplicon sequencing, and genetic analysis in the cecum content were carried out by the Genedenovo Biotechnology Company (Guangzhou, China). In order to analyze the gut microbiome in the cecum content, we used 16S sequencing technology. The V3 + V4 domains of 16S rDNA were amplified by barcoded specific primers 341F (CCTACGGGNGGCWGCAG) and 806R (GGACTACHVGGGTATTCTAAT). Qubit 3.0 fluorometer was used for recovery and quantitative amplification, and a sequencing library was constructed according to the official Illumina information. Finally, sequencing was completed using the PE250 model in the Hiseq2500 system.

### 2.9. Statistical Analysis

Statistical analysis was performed using Origin software (Origin 2018, OriginLab, Northampton, MA, USA). Data were normally presented as the mean ± standard error of the mean (SEM). Difference analyses of the results were conducted by the one-way ANOVA test with Fisher LSD multiple comparisons, while statistical significance was determined as *p <* 0.05. Letters a, b, and c were applied in all histograms, indicating significant differences between groups. A gut microbiota analysis was conducted using Omicsmart with the online platform (http://www.omicsmart.com accessed on 13 December 2021).

## 3. Results

### 3.1. Dietary 2′-FL Supplementation Reduces Systemic Oxidative Stress Damage

The levels of SOD, CAT, GSH, GSH-PX, and MDA in the serum were measured for the effects of 2′-FL in the d-gal-induced aging model. As shown in Figure 1, the level of these antioxidant enzymes (SOD, CAT, GSH, and GSH-PX) were significantly decreased (*p <* 0.05), and MDA was significantly increased (*p <* 0.05) in the d-gal group compared with the control group. The dietary intervention of 2′-FL increased the level of antioxidant enzymes in the blood (*p <* 0.05), reducing the oxygen free radicals produced by the enzyme system and improving the antioxidant capacity. Otherwise, MDA was a product of lipid peroxidation induced by oxygen free radicals, promoting lipid oxidation and forming a vicious cycle. The 2′-FL intervention significantly reduced the production of MDA in the blood (*p <* 0.05).

### 3.2. Dietary 2′-FL Supplementation Ameliorates Aging-Related Gut Mucosal Barrier Impairment

The gut mucosal layer was stained by immunocytochemistry to observe the intervention effect of 2′-FL on gut barrier repair in aging models. As shown in Figure 2a, the thickness of the mucus layer in the d-gal group was lowest among the groups, with extremely scarce mucin in the goblet cells. All of them indicated that aging was positively associated with the decline of the gut mucosal barrier function. Furthermore, the gut mucosal barrier-related genes expression was determined to measure the defense functions. In the d-gal group, the expression level of MUC2, Claudin-1, and E-cadherin significantly decreased (*p <* 0.05) compared with the control group. The expression level of the gut mucosal barrier-related genes in the 2′-FL intervention group increased compared with the d-gal group (*p <* 0.05), but there was still a gap with the control group level (Figure 2b). All these data suggested that the 2′-FL intervention repaired the damage of the barrier in the d-gal model.

### 3.3. Dietary 2′-FL Supplementation Ameliorates Aging-Related Inflammation and Regulate Nrf2 Signaling Pathway

The relative mRNA expression of inflammatory cytokines in the colon was measured to evaluate the effect of the 2′-FL intervention on inflammation. As shown in Figure 3a, the expression level of TNF-α and IL-1β in the d-gal group showed a significant increase (*p <* 0.05) compared with the control group. When intervened by 2′-FL, the expression level of TNF-α and IL-1β significantly decreased (*p <* 0.05). The expression of IL-1β had no difference between the control group and the 2′-FL intervention group, but TNF-α was still a little higher than the control group. These data indicated that the 2′-FL intervention effectively reduced the inflammatory response in the d-gal model.

Compared with the control group, d-gal treatment significantly decreased the expression levels of Nrf2, NQO1, and HO-1 (*p <* 0.05) but increased the expression level of Keap1 (*p <* 0.05). However, 2′-FL could reverse this situation; the relative gene expression levels of Nrf2, NQO1, and HO-1 were all significantly increased (*p <* 0.05), while the level of Keap1 was significantly decreased (*p <* 0.05) (Figure 3b).

### 3.4. Dietary 2′-FL Supplementation Has Effects on the AMPK/SIRT1/FOXO1 Longevity Pathway

qRT-PCR and Western blotting were applied to determine the AMPK/SIRT1/FOXO1 longevity-related signaling pathway in the colon for the mechanism of 2′-FL intervention on aging. As shown in Figure 4a, compared with the control group, the relative mRNA expression levels of SIRT1, PGC-1α, and FOXO1 in the d-gal group were significantly reduced (*p <* 0.05). Compared with the d-gal group, the 2′-FL intervention group significantly upregulated the expression levels of these genes (*p <* 0.05), which were not significantly different from the control group. Furthermore, the expression of AMPK, pAMPK, SIRT1, and FOXO1 were determined at the protein level. These results showed that the protein levels of pAMPK, SIRT1, and FOXO1 in the 2′-FL intervention group were significantly upregulated compared with the d-gal group, but there was no significant difference from the control group on pAMPK and SIRT1. These data indicate that the 2′-FL intervention has effects on inhibiting the process of aging through the AMPK/SIRT1/FOXO1 signaling pathway.

The AMPK pathway could be activated by G-protein-coupled receptors (GPRs). We believe that 2′-FL may bind to receptor members of the GPR family through its metabolite SCFAs, thereby activating the downstream AMPK signaling pathway. We measured the expression of GPR41 and GPR43, which were main receptors of SCFAs in the colon. As shown in Figure 4b, compared with the d-gal group, although the expression level of GPR41 increased, there was no significant difference. The GPR43 expression level of the 2′-FL intervention group was significantly upregulated compared to the d-gal group (*p <* 0.05) but not different from the control. To verify the changes in the GPRs, we further measured the content of SCFAs in the colon contents. We found that the contents of acetic acid, propionic acid, and butyric acid in the 2′-FL intervention group were highest among the groups (*p <* 0.05), while the content of valeric acid was not different in all the groups (Figure 4c). These data further proved that 2′-FL intervention could regulate the SIRT1-related pathway, and one of the reasons was achieved through GPRs.

### 3.5. Effects of Dietary 2′-FL Supplementation on Gut Microbiota in Aging Mice

16S sequencing technology was applied to measure the species differences of the gut microbiota on 2′-FL intervention. At the phylum level, the relative abundance of *Firmicutes* and *Deferribacteres* in the d-gal treatment group increased, but the relative abundance of *Verrucomicrobia* decreased. After a long-term 2′-FL intervention, the relative abundance of *Verrucomicrobia* reversed dramatically compared with d-gal. The proportion of *Verrucomicrobia* was 0.89% in the d-gal treatment group and 3.5% in the 2′-FL intervention group. In addition, the ratio of *Firmicutes/Bacteroidetes* also increased significantly (Figure 5a). At the genus level, the relative abundances of *Lachnospiraceae_NK4A136_group*, *Oscillibacter*, *Prevotellaceae_UCG-001*, and *Akkermansia* in the d-gal treatment group were significantly reduced (*p* < 0.05), while the relative abundances of *Ruminiclostridium* and *Ruminiclostridium_9* were extremely significantly increased (*p* < 0.01). Surprisingly, the 2′-FL intervention significantly reversed the changes. The relative abundance of *Akkermansia* in the 2′-FL intervention group significantly increased and restored its original abundance, and the relative abundance of *Ruminiclostridium_9* and *Butyricimonas* was significantly reduced (*p* < 0.05) (Figure 5b). The dilution curve could be used to measure the diversity of species directly, and the height of the curve reflected the abundance of the species. As shown in Figure 5c, the control group had the highest species abundance, followed by the d-gal treatment group, while the 2′-FL intervention group had the lowest species abundance among all the groups. Querying the SILVA database, we predicted the function of the gut microbiota. As shown in Figure 5d, compared with the control group, the function of signal transduction in the d-gal treatment group was significantly reduced (*p* < 0.01), while the expression of cancer-related pathways was significantly upregulated (*p* < 0.05). The 2′-FL intervention group significantly increased the expression of translation function (*p* < 0.05) and decreased the expression of neurodegenerative disease-related pathways compared with the d-gal treatment group (*p* < 0.01).

Moreover, the Pearson correlation analysis was used to investigate the relevance level between the gut microbiota and environmental factors (TNF-α, IL-1β, MUC2, Claudin1, E-cadherin, SIRT1, PGC-1α, and FOXO1). At the genus level (Figure 5d), *Akkermansia* was positively correlated with the expression of longevity genes SIRT1 and FOXO1 but negatively correlated with IL-1β. Some unidentified bacteria such as *Ruminiclostridium* and *Ruminiclostridium_9* were negatively correlated with the expression of gut mucosal barrier-related genes (MUC2, Claudin1, and E-cadherin) but positively correlated with inflammatory cytokines (TNF-α). At the species level (Figure 5f), *Akkermansia*_*muciniphila* was positively correlated with the expression of SIRT1 and FOXO1 but negatively correlated with IL-1β, which was consistent with the results at the genus level. *Lachnospiraceae_bacterium_615*, *Lachnospiraceae_bacterium_DW17*, *Clostridium_sp_Marseille-P2776*, and *Bacteroides_acidifaciens* were negatively correlated with the expression of the gut mucosal barrier-related genes (MUC2, Claudin1, and E-cadherin). In addition, *Lachnospiraceae_bacterium_DW17* positively correlated with inflammatory cytokines (TNF-α and IL-1β). These data supported the conclusions that the interaction of 2′-FL with the gut microbiota had a positive effect, increasing the abundance of beneficial bacteria like *Akkermansia* and regulating the expression of genes related to inflammation and antiaging.

### 3.6. Effects of Dietary 2′-FL Supplementation on Brain Tissue Injury in Aging Mice

Aging would reduce cone cells and inflammatory infiltration in the hippocampus of the host’s brain, thereby promoting the production of inflammatory cytokines and oxidative stress products, ultimately leading to a series of degenerative changes such as neurotransmitter metabolism disorders. As shown in Figure 6a, the 2′-FL intervention significantly reduced cone-cell cytoplasmic Nissl bodies and some cytopathic phenomena, indicating that the 2′-FL intervention could effectively alleviate brain damage caused by d-gal. In addition, we measured the expression levels of apoptosis-related genes in the brain. As shown in Figure 6b, the expression levels of Bax and caspase-3 in the 2′-FL intervention group were not significantly different from those in the control group but were significantly lower than in the d-gal group (*p <* 0.05). The expression level of Bcl-2 in the 2′-FL intervention group was the highest among the groups (*p <* 0.05).

As metabolites of 2′-FL, SCFAs can cross the blood–brain barrier and affect brain functions. In order to verify their effect on the brain, we used qRT-PCR to determine the genes expression of the SIRT1-related pathway in the brain. The 2′-FL intervention group significantly increased the expression of SIRT1 and FOXO1 (*p <* 0.05) but was not different from the control group in the SIRT1 level (Figure 6c).

As well as the intestine, we also measured the Nrf2 signaling pathway in the brain. Compared with the control group, the expression of Nrf2, HO-1, and NQO-1 in the d-gal group all had a decreasing trend. However, only the decrease in the expression of NQO-1 was significant (*p <* 0.05), which showed that d-gal had little effect on the Nrf2 signaling pathway in the brain. Compared with the d-gal group, the 2′-FL intervention group significantly increased (*p <* 0.05) the levels of Nrf2 and NQO-1 (Appendix A). These results indicated that the effect of 2′-FL intervention on the brain was not significantly related to the Nrf2 signaling pathway.

## 4. Discussion

Aging and various diseases caused by aging are health problems focused on the whole world. Currently, the prevention and treatment strategies are still insufficient [34]. Aging is a complex multifactor process related to apoptosis, oxidative stress, autophagy, and inflammation. The imbalance between the production and elimination of ROS in the body was reported to mainly cause aging. Moreover, the excessive accumulation of ROS leads to tissue damage and immune dysfunction [35,36]. Therefore, applying antioxidants to maintain the balance of the ROS response is a promising prevention and treatment strategy.

2′-FL is a natural oligosaccharide present in breast milk, which has a variety of effects on the health of humans. The functions of anti-inflammation, antioxidation, and brain development have been confirmed in 2′-FL [23], but few reports have shown its effects on aging. In this study, the d-gal accelerated aging model was used to investigate the repair effect of the 2′-FL intervention on the intestinal and brain injuries of aging mice. After 2′-FL is taken into the intestine, part of it is used by the gut microbiota and metabolized into SCFAs and other small molecules, and another part interacts with the intestinal epithelial cells to cause a series of changes in the body. Additionally, a tiny part is absorbed into the periphery loop [37,38]. We hypothesized that the 2′-FL intervention would change the composition of the gut microbiota, as well as abundance, but also, its metabolites would have better effects on the body. These effects may improve the health of the intestine and brain in the aging situation.

The existing literature has proven that antioxidant enzymes can scavenge free radicals and resist oxidative stress injury caused by ROS [39]. In this study, the levels of oxidative stress-related substances in the serum were measured to evaluate the systemic antioxidant function of 2′-FL intervention. A series of reactions could catalyze free radicals in the body by antioxidant enzymes (SOD, CAT, GSH, and GSH-PX) to produce water and oxygen, thereby reducing injury [18]. MDA is a product of lipid peroxidation, one of the potential factors of DNA damage. Studies have reported that the activity of MDA increased during aging [40]. In this study, the 2′-FL intervention significantly reversed this consequence, increasing the activity of antioxidant enzymes and decreasing the MDA activity. These results indicated that the supplementation of 2′-FL had a protective effect on oxidative stress.

Nrf2 regulates cellular oxidative stress and induces the antioxidant responses, such as regulating redox balance, energy metabolism, autophagy, DNA repair, and mitochondrial physiological functions [41]. Under normal physiological conditions, Nrf2 and Keap1 are combined in the cytoplasm and are inactive. If this state is maintained long enough, Nrf2 will eventually be ubiquitinated and degraded [42]. However, when stimulated by oxidative stress and other stimulus, the binding of the Nrf2/Keap1 complex will be interrupted. Then, Nrf2 will be released and transferred into the nucleus, where it binds with ARE and activates downstream genes such as HO-1 and NQO-1 [43]. The current research has shown that the Nrf2 signaling pathway is closely related to aging and neurodegenerative diseases [44,45]. For example, studies have reported that trehalose relieved the oxidative damage potentially by regulating the Nrf2 pathway in a d-gal model of mice [46]. In addition, downregulation of the Nrf2 pathway was also found in C57BL/6J mice kidneys with d-gal-induced senescence that was intervened by seaweed oligosaccharides [47]. Our research results showed that the supplementation of 2′-FL could effectively activate the Nrf2 signaling pathway by reducing the transcription level of Keap1 and enhancing the transcription level of Nrf2 in the colon, thereby promoting the expression of HO-1 and NQO1 in aging model mice that were downstream target genes of Nrf2. However, the activation of this signaling pathway has not been detected in the brain. It is speculated that the reason may be that 2′-FL itself cannot pass the blood–brain barrier, and the contents of its metabolites that an act on the brain are too small after circulation to fully activate the Nrf2 signaling pathway.

Inflammation is one of the vital factors that lead to aging. In this regard, TNF-α and IL-1β are classic cytokines that mediate immune and inflammatory responses whose components are small molecule-soluble proteins. TNF-α is produced in macrophages and works through two receptors (TNFR1 and TNFR2), leading to tissue damage and cell apoptosis [48]. IL-1β is a central regulator of inflammation and the immune response, released by inflammasomes activating macrophages. IL-1β is related to the pathogenesis of neurodegenerative diseases and other diseases [49]. Our research results showed that 2′-FL supplementation reduced the gene expression level of TNF-α and IL-1β. From this point of view, 2′-FL might attenuate inflammation, thereby protecting the intestinal inflammatory injury in the d-gal-induced model.

The intestinal barrier integrity can be destroyed by inflammation during the aging process [50]. In the *E. coli* O157 infection model in vivo, the 2′-FL intervention significantly increased the expression levels of MUC2 and tight junction protein occludin [31]. Moreover, scholars have constructed Caco-2 and HT29-MTX cell models induced by TNF-α and IFN-γ cytokines in vitro, discovering that 2′-FL had a great significant protective effect on the intestinal epithelial barrier among different human milk oligosaccharides. Furthermore, it is speculated that 2′-FL is the most effective single HMO to prevent the increase of epithelial permeability after inflammation induction [51]. These studies showed the positive role of 2′-FL in protecting the gut mucosal barrier. In addition, it was reported that Nrf2 knockdown could inhibit the resveratrol-induced upregulation of antioxidant and intestinal epithelial cell barrier genes, indicating that Nrf2 might be related to intestinal barrier damage caused by oxidative stress [52]. In this study, intervened by 2′-FL in the aging process, the gene expression levels of MUC2 and the tight junction proteins (Claudin1 and E-cadherin) were significantly upregulated, which was consistent with the reduction of inflammatory cytokines and the upregulation of the Nrf2 signal pathway.

The dynamic balance of the gastrointestinal tract is generally maintained by the gut microbiota. The composition of the gut microbiota, as well as abundance, would change differently during different life periods. With the increase of age, the abundance of *Firmicutes/Bacteroides* and *Akkermansia* in the intestine decreases, whereas the abundance of those harmful bacteria such as *Proteus*, *Enterobacter*, and *Enterococcus* increases [53]. It has been determined that there are more bacteria called *Akkermansia* in thin people. This has led to claims that this bacterium can prevent weight gain. One of the foods that will feed the bacteria is red grapes. During the aging process, these changes in the gut microbiota can cause a series of diseases. In this study, the 2′-FL intervention significantly increased the abundance of *Verrucomicrobia*, while *Akkermansia* contributed the most at the genus level. Furthermore, the correlation analysis of the environmental factors proved that *Akkermansia* is positively correlated with the SIRT1-related longevity signaling pathway. *Akkermansia* is one of the few known species in the *Verrucomicrobia* phylum that can degrade mucin into acetic and propionic acids, further promoting intestinal integrity [54]. In a study of fecal bacteria transplantation in premature aging mice, *Akkermansia* was found to extend the lifespan of premature aging mice. Moreover, in a cohort study, it was also found that *Verrucobacteria* in the gut microbiota of centenarians increased, while the *proteobacteria* decreased, indicating that *Akkermansia* had the potential to extend the lifespan [55]. In addition, we observed that the abundance of *Butyricimonas* in the 2′-FL intervention group was decreased. However, there are reports that one species called *Butyricimonas virosa* under *Butyricimonas* is a harmful bacterium that can cause bacteremia [56]. It is worth noting that *Lachnospiraceae_bacterium_615* and *Lachnospiraceae_bacterium_DW17* under *Lachnospiraceae* have a negative correlation with the intestinal barrier. Although members of the *Laospirillaceae* family are the main producers of SCFAs, the different taxa of the *Laospirillaceae* are also associated with different enteral and extraintestinal diseases. Specific taxa of the *Laospirillaceae* family were involved in metabolic syndrome, obesity, diabetes, liver disease, and inflammatory bowel disease [57]. These observations suggested that 2′-FL exerted benefits potentially by regulating the gut microbiota in aging.

AMPK plays a vital role in regulating cell energy homeostasis and preventing aging [58]. As a NAD^+^-dependent deacetylase, SIRT1 can regulate the AMPK signaling pathway through the deacetylation of certain key proteins (such as PGC-1α [59], etc.), thereby delaying cell senescence. In addition, SIRT1 can also interact with a variety of other senescence signaling molecules, such as FOXO1, PGC-1α, NF-κB, p53, and Bax, thereby regulating oxidative stress, inflammation, autophagy, and apoptosis [8,60,61]. In this study, the expressions of SIRT1, FOXO1, and PGC-1α in the 2′-FL intervention group were upregulated in the colon, and the expressions of SIRT1 and FOXO1 in the 2′-FL intervention group were upregulated in the brain. These results indicated that the supplementation of 2′-FL might reduce the damage associated with aging through the AMPK/SIRT1/FOXO1 signaling pathway.

In the colon, 2′-FL is metabolized by bacteria into a variety of SCFAs [38]. SCFAs can provide energy for colonic epithelial cells, maintain intestinal barrier functions, and regulate immune responses [62]. According to research reports, aging is closely related to the decrease of the SCFA content in the intestine [63]. Multiple evidence indicates that SCFAs can directly or indirectly activate AMPK [64,65], but the mechanism by which SCFAs activate AMPK is still unclear. GPR41 and GPR43 are of the G-protein-coupled receptors family, which are the main receptors for SCFAs. We measured the gene expression levels of GPR41 and GPR43 in the colon and found that the expression level of GPR43 was upregulated by 2′-FL intervention. Based on those results, we speculated that the regulatory mechanism of 2′-FL intervention on the AMPK/SIRT1/FOXO1 signaling pathway might be mediated by GPRs.

Aging is always accompanied by the deterioration of brain function, such as oxidative stress damage, cognitive impairment, increased apoptosis, and neurodegenerative diseases. In a study of AlCl_3_-induced cognitive impairment in rats, a lycopene intervention was considered to inhibit the process of apoptosis, which was accompanied by a weakness of the hippocampal lesions [66]. Bcl-2 is one of the antiapoptotic members, while Bax is a proapoptotic member, both of which are proteins related to apoptosis. The imbalance between Bcl-2 and Bax could induce mitochondrial membrane permeabilization, causing mitochondria to release cytochrome c and activate the caspase cascade [67]. The caspase cascade is the main process of cell apoptosis, where the most critical enzyme is caspase-3 [68]. As mentioned earlier, the Nrf2 signaling pathway in the brain did not change significantly. We continued to measure the expression level of apoptosis-related genes. These research results showed that the supplementation of 2′-FL significantly inhibited apoptosis in the brain by regulating the expression of apoptosis-related genes. From this point of view, 2′-FL had a potential alleviating effect on brain dysfunction caused by d-gal.

In conclusion, 2′-FL was beneficial in oxidative stress, the gut mucosal barrier, and inflammation in the d-gal-induced aging model. Its mechanism depended on the regulation of the AMPK/SIRT1/FOXO1 and Nrf2/Keap1 signaling pathways, respectively. 2′-FL had a relieving effect on brain damage, effectively inhibiting the apoptosis of brain tissue cells. These results indicated that 2′-FL and its metabolites had a regulatory effect on the homeostasis of the gut environment and a potential protective effect on the brain. This study clarified that 2′-FL supplementation was beneficial in the aging process, providing a new idea for comprehending the function of 2′-FL.

## Figures and Tables

**Figure 1 foods-11-00151-f001:**
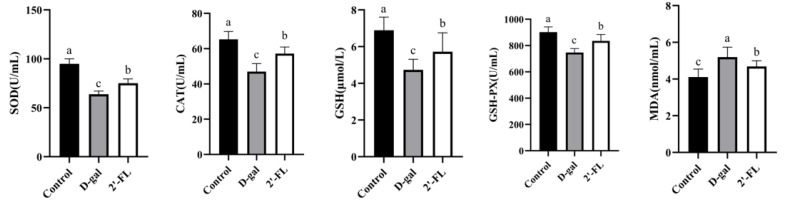
Effects of 2′-fucosyllactose (2′-FL) on oxidative stress. Letters a, b, and c indicated significant differences between groups (*p* < 0.05).

**Figure 2 foods-11-00151-f002:**
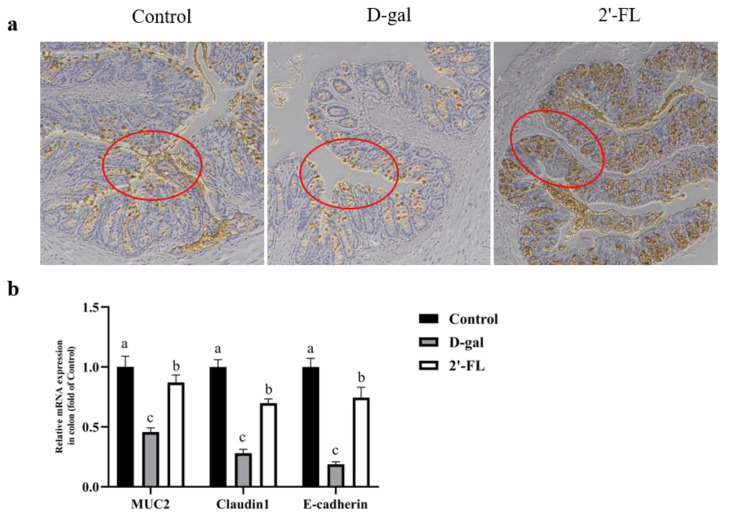
Effects of 2′-FL on the gut mucosal barrier. (**a**) Immunohistochemistry of the mucous layer (MUC2, zoom in 100×; circles indicate the expression of MUC2). (**b**) Expression of mucoprotein and tight junction proteins at the gene level. Letters a, b, and c indicated significant differences between groups (*p* < 0.05).

**Figure 3 foods-11-00151-f003:**
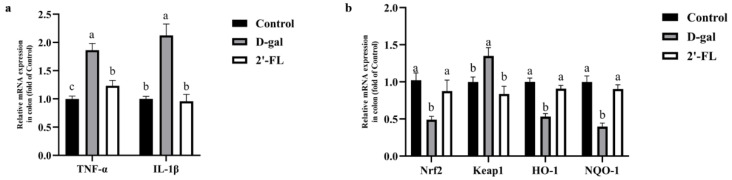
Effects of 2′-FL on inflammation and the nuclear factor E2-related factor 2 (Nrf2) signaling pathway in the colon. (**a**) Expression of inflammatory cytokines at the gene level. (**b**) Expression of the Nrf2 signaling pathway at the gene level. Letters a, b, and c indicated significant differences between groups (*p* < 0.05).

**Figure 4 foods-11-00151-f004:**
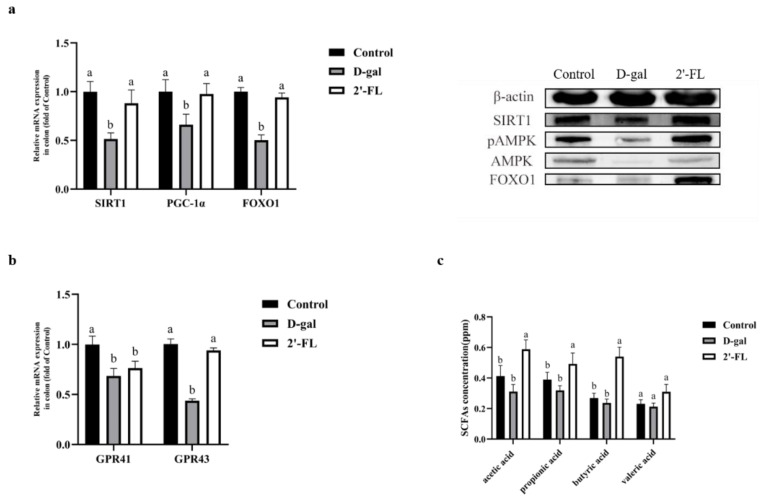
Effects of 2′-FL on longevity pathways. (**a**) Gene and protein expressions of the AMPK/SIRT1/FOXO1 pathway in the colon. (**b**) Expression of G-protein-coupled receptors (GPRs) in the colon at the gene level. (**c**) Content of short-chain fatty acids (SCFAs) in the colon contents. Letters a and b indicated significant differences between groups (*p* < 0.05).

**Figure 5 foods-11-00151-f005:**
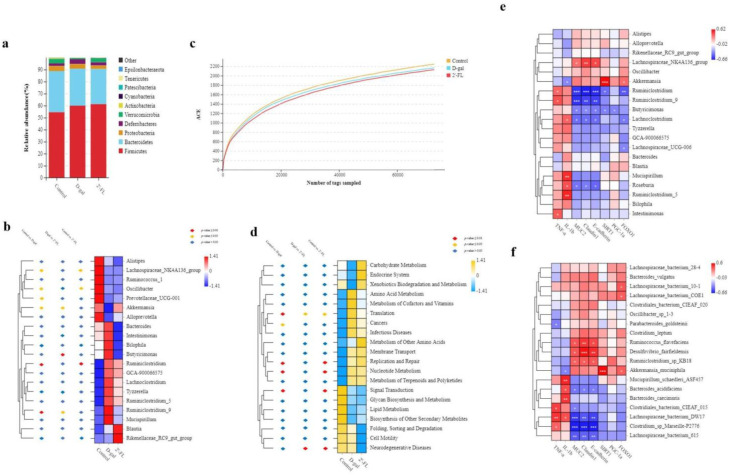
Analysis of the gut microbiota in all three groups. (**a**) The level of phylum. (**b**) The level of genus. (**c**) Dilution curve. (**d**) Tax4Fun analysis in level 2 based on the SILVA database. (**e**) Pearson analysis at the genus level. * *p* < 0.05, ** *p* < 0.01, and *** *p* < 0.001. (**f**) Pearson analysis at the species level. * *p* < 0.05, ** *p* < 0.01, and *** *p* < 0.001.

**Figure 6 foods-11-00151-f006:**
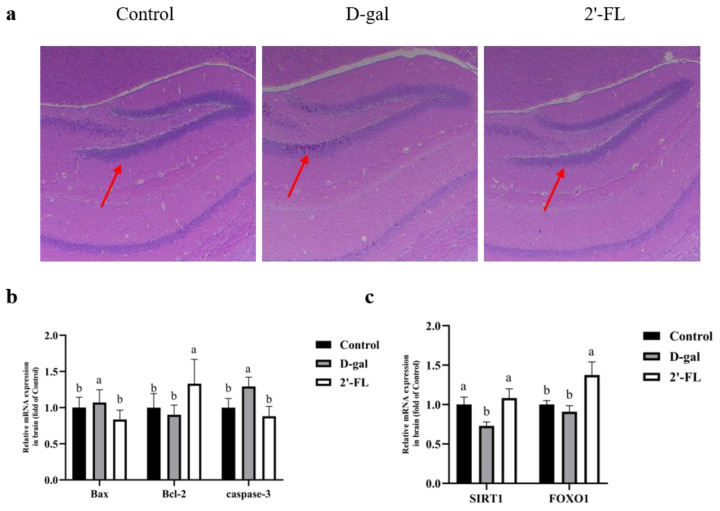
Effects of 2′-FL on inflammation, longevity, and apoptosis in the brain. (**a**) H&E staining of the hippocampus (zoom in 100×; arrows indicate inflammatory infiltration). (**b**) Expression of apoptosis at the gene level. (**c**) Expression of the SIRT1/FOXO1 pathway at the gene level. Letters a and b indicated significant differences between groups (*p* < 0.05).

## Data Availability

The datasets generated for this study are available on request to the corresponding author.

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
