# Peer review of "2′-Fucosyllactose Ameliorates Oxidative Stress Damage in d-Galactose-Induced Aging Mice by Regulating Gut Microbiota and AMPK/SIRT1/FOXO1 Pathway"

_foods, 2022, doi:10.3390/foods11020151_

Round 1

Reviewer 1 Report

The manuscript deals with the effect of 2'-Fucosyllactose on Gut Microbiota resulting in the suppression of oxidative damages of cells and the aging process. The manuscript is well written and the objective of the study is clear. Justification is also clearly stated. However, the authors have to rectify the following issues… 

  • Experimental Procedures of Animal section is vague. Rewrite the animal treatment method and include the institutional animal ethical committee approval number.
  • Why did the authors select all animals are male-only? What are the direct correlates of male animals with the aging process?
  • Kindly explain the changes in the immunohistochemistry the Figures 2 and 6 (a) by indicating the specific region in the figures.
  • Please check the figure 2 (B) Y axis’s title.
  • Please check and change the supplementary table title (Table 1. This is a table. Tables should be placed in the main text near to the first time they are cited.)
  • What are the primary and secondary antibodies used in this study please specify it. (with the primary antibody. The mixture was incubated…….. secondary antibody, followed by adding developers to develop……)
  • The English language must be revised. Kindly check for all typos and grammatical errors (example: compared with the Con group.)
  • In figure 2 (a) the level of beta-actin in D-gal and 2-FL appears to be higher than the control why?
  • In figure 5 (b) and (e) what is GCA-900066575 please explain?
  • Please rotate figure 5 (a) vertically for better readability.
  • Kindly explain the results of microbiome analysis in detail, if possible discuss with the genus and species level of correlations.

Reviewer 2 Report

1- The information that the D-galactose-induced accelerated aging model will be applied is given at the end of the Introduction. (Line 108)

I think it would be more informative for those who do not know the literature if this model was explained as I have stated below.

Injections of low-dose D-galactose into mice were first reported in China for signs of accelerated aging. Neurological damage, decreased antioxidant enzyme activity, and weakened immune response have been demonstrated in this model of aging. D-gal is a reducing sugar that reacts with free amino groups of protein and peptide amino acids to form advanced glycation end products. These biochemical processes mentioned cause oxidative stress and cellular damage. D-gal is converted to galactitol, which can produce reactive oxygen species in the cell and stimulate osmotic stress, as well as accumulate in the cell.

2-Descriptive information about 2'-Fucosyllactose could have been given more comprehensively.

2'-Fucosyllactose (2'-FL) is a type of neutral, low molecular weight, human milk oligosaccharide (HMO) recognized to play an extremely important role in host immunity. The unique nature of 2'-FL allows it to resist digestion in the gastrointestinal tract, with very little entering the blood and systemic circulation, causing the majority to easily reach the large intestine for microbial decomposition.

  1. It is not specified whether the mice used in the study were born by cesarean or normal route. This may be important as it affects the gut microbiota. (Line 279)
  2. An important finding is that the 2'-FL intervention repairs the damage of the barrier in the D-gal model. (Line 202-203-218-233-234-325-326)
  3. With the increase of age, the abundance of Firmicutes/Bacteroides and Akkermansia in the intestine decreases, whereas the abundance of those harmful bacteria such as Proteus, Enterobacter, and Enterococcus increases. (Line 429-430) It has been determined that there are more bacteria called Akkermansia in thin people. This has led to claims that this bacterium can prevent weight gain. One of the foods that will feed this bacteria is red grapes. By adding this information, the relationship between grape consumption, resveratrol content and aging could be explained. (Line 419-423)
  4. Thank you to those who contributed to the valuable work. Best wishes.

Reviewer 3 Report

the paper is well written and has a very interesting framework. but in my opinion, there is a few places for improvement. the certainty of analytical methods for fatty acid. the second remark is about the designation of SCFA if we look at IUPAC fatty acid series starting from butyric acid, with the clarification that extended description summarizes all aliphatic carboxylic acid. I suggest authors put references and describe why they decide to put acetic and propionic acid in fatty acid series.
